# Oxidative Stress in Type 2 Diabetes: The Case for Future Pediatric Redoxomics Studies

**DOI:** 10.3390/antiox11071336

**Published:** 2022-07-07

**Authors:** Stephanie N. Alu, Evan A. Los, George A. Ford, William L. Stone

**Affiliations:** Department of Pediatrics, Quillen College of Medicine, Johnson City, TN 37614, USA; alusn1@etsu.edu (S.N.A.); losea1@etsu.edu (E.A.L.); fordga@etsu.edu (G.A.F.)

**Keywords:** redoxomics, diabetes, oxidative stress, antioxidants, insulin resistance, skeletal muscle, glucose

## Abstract

Considerable evidence supports the role of oxidative stress in adult type 2 diabetes (T2D). Due to increasing rates of pediatric obesity, lack of physical activity, and consumption of excess food calories, it is projected that the number of children living with insulin resistance, prediabetes, and T2D will markedly increase with enormous worldwide economic costs. Understanding the factors contributing to oxidative stress and T2D risk may help develop optimal early intervention strategies. Evidence suggests that oxidative stress, triggered by excess dietary fat consumption, causes excess mitochondrial hydrogen peroxide emission in skeletal muscle, alters redox status, and promotes insulin resistance leading to T2D. The pathophysiological events arising from excess calorie-induced mitochondrial reactive oxygen species production are complex and not yet investigated in children. Systems medicine is an integrative approach leveraging conventional medical information and environmental factors with data obtained from “omics” technologies such as genomics, proteomics, and metabolomics. In adults with T2D, systems medicine shows promise in risk assessment and predicting drug response. Redoxomics is a branch of systems medicine focusing on “omics” data related to redox status. Systems medicine with a complementary emphasis on redoxomics can potentially optimize future healthcare strategies for adults and children with T2D.

## 1. Introduction

About one in ten, or 37 million Americans, have diabetes mellitus (DM), with an annual estimated cost (in 2017) of $237 billion [1,2]. About 90–95% of Americans with DM have type 2 (T2D), which usually develops after age 45 but is rapidly increasing in pediatric patients as young as 4 years old. The overall rate of T2D in the U.S. is 8.1 per 100,000 in children aged 10–14 and 11.8 per 100,000 in adolescents aged 15–19. The highest rates are among Native American Indians [3,4]. However, the most startling statistic is that 1 in 5 adolescents are currently living with prediabetes [5]. Prediabetes dramatically increases the risk of developing T2D and other complications such as kidney or heart disease [5,6]. The projected prevalence of T2D in the pediatric population is estimated to increase by 49% over the next few decades [7]. Earlier development of T2D in youths will likely result in diabetes-related complications occurring at a younger age with a decreased life expectancy and markedly increased healthcare costs.

As outlined in Figure 1, it is now generally accepted that T2D progresses in four stages, i.e., insulin resistance, prediabetes, T2D, and T2D with vascular complications [8,9]. In keeping with our focus on pediatric T2D, we will emphasize the first three stages and the potential etiological role of oxidative stress. As detailed below, dietary-induced oxidative stress is a likely early initiating factor for insulin resistance and, therefore, particularly important in childhood progression to T2D. Early intervention to prevent dietary-induced oxidative stress may help delay or prevent the progression to T2D [10]. The past, current, and future role of oxidative stress in T2D will be summarized. A future systems medicine approach (detailed in Section 3), with a complementary emphasis on redoxomics, may provide early risk assessment and help guide healthcare strategies. Redoxomics, with its comprehensive focus on oxidative stress-health risk factors, holds promise for preventing, treating, and monitoring T2D [11].

### 1.1. Obesity, Lack of Physical Activity, Poor Diet, and Insulin Resistance Are Major Risk Factors for Pediatric T2D

Major risk factors for pediatric T2D include overweight, obesity, a lack of physical activity, a poor diet, and insulin resistance [4,12]. Moreover, pediatric T2D is associated with increased plasma levels of free fatty acids (FFAs), primarily caused by an increased fat mass [13]. Frohnert et al. [13] found that the relationship between FFAs and insulin resistance is not significant in early adolescence but becomes so in young adulthood.

### 1.2. The Role of Postprandial Oxidative Stress (POS) in T2D

In a prescient 2006 article, Wright et al. reviewed the evidence supporting the role of postprandial oxidative stress (POS) as a “root cause” of T2D [14]. These authors summarized the links between excess dietary calories and POS and the resulting insulin resistance and eventual beta-cell dysfunction. They also emphasized the potential role of postprandial hyperglycemia in promoting the formation of plasma-advanced glycation end products (AGEs), increasing oxidative stress and inflammation and reducing nitric oxide (NO) availability. Since 2006 there has been considerable new evidence supporting the role of POS in T2D [15]. We now know that excess dietary fat is much more significant than appreciated in 2006. As detailed below, excess dietary fat promotes skeletal muscle mitochondrial-induced oxidative stress leading to insulin resistance [16]. As indicated in Figure 1, plasma-POS likely reflects both dietary pro-oxidant/antioxidant factors as well as skeletal muscle mitochondrial-induced oxidative stress and dietary AGEs [17,18,19].

### 1.3. Dietary Fat Level Is a Key Determinant of Plasma-POS

Direct observations in human plasma samples indicate that shortly after food ingestion, there is a transient increase in the production of reactive oxygen species (ROS) and a simultaneous decrease in antioxidants [20]. Bloomer et al. compared plasma POS in healthy young male adults after consuming isocaloric amounts of carbohydrate (dextrose) or lipid (heavy whipping cream) [20]. Dietary fat generated more POS than dietary carbohydrates. As expected, blood glucose significantly increased after the carbohydrate meals but not after the lipid meals. Sottero et al. suggest that POS is exacerbated in T2D due to AGEs and increased plasma levels of FFAs (aka hypernefemia) [15].

### 1.4. Dietary Fat Type and Dietary Antioxidants Influence Plasma-POS

The type of dietary fat has also been shown to play a role in plasma POS with a high level of monounsaturated fatty acid (MUFA) causing less oxidative stress than a high level of saturated fatty acid (SFA) [21,22]. The MUFA content of the Mediterranean diet is high and, when compared to a diet high in SFA, improves POS, nitric oxide (NO) levels, and capillary flow in the elderly [22]. Recent evidence shows the Mediterranean diet to be beneficial for T2D and prediabetes [23,24]. The Mediterranean diet, which is also high in polyphenolic antioxidants, increases endogenous antioxidant activity and decreases pro-oxidant activity [25]. In obese children or adolescents, a Mediterranean diet improves body mass index (BMI) as well as glucose and lipid profiles [26]. Supplementing a high SFA meal with a pool of antioxidants reduces POS and improves endothelial function in both normal and insulin-resistant subjects [17].

The measurement of plasma POS parameters in the pediatric population may prove to be of clinical relevance. As indicated in Figure 1, excess dietary fat-induced mitochondrial oxidative stress in skeletal muscle is an underlying initial cause of insulin resistance. We will next summarize the pathophysiology of T2D, emphasizing the role of oxidative stress. The key molecular players discussed below are also probable candidates to be measured in future pediatric redoxomics studies.

## 2. The Pathophysiology of T2D

T2D is a complex disease involving multiple organ systems with numerous alterations in carbohydrate, protein, and lipid biochemistry. This complexity continues at the cellular level with the involvement of multiple subcellular organelles. A comprehensive review of the pathophysiological mechanisms underlying T2D is beyond the scope of this review. A concise summary of oxidative stress, ROS, and antioxidants can be found online [27]. Given the importance of glucose and insulin in T2D, we will briefly outline their key normal and pathophysiological roles [28]. Intracellular glucose oxidation is a primary source of energy for producing ATP. Energy metabolism, in turn, is highly linked to the production of ROS [29].

Glucose transport from the circulation into tissues is primarily governed by four facilitative transporters, i.e., GLUT1, GLUT2, GLUT3, and GLUT4, each with a physiologically relevant K_m_ (Michaelis constant) and V_max_ (reaction rate when the transporter/enzyme is fully saturated by substrate). GLUT1 is highly expressed on red blood cells, is insulin-independent, and rapidly equilibrates RBC cytoplasmic glucose with plasma glucose. Endothelial cells also rely on GLUT1 for glucose transport [30]. GLUT4 is an insulin-responsive transporter and is present in the cells of skeletal muscle, heart, liver, and adipose tissue. Skeletal muscle primarily utilizes GLUT4 but may also express low levels of GLUT1 [31,32]. Insulin release from the beta-cells of the pancreatic islets of Langerhans is normally triggered by a postprandial increase in plasma glucose. Insulin binding to the insulin receptor (IR) on sensitive cells results in the activation (by phosphorylation) of the AKT protein, which is a serine/threonine-specific protein kinase. AKT phosphorylation, in turn, results in the translocation of GLUT4 to the plasma membrane and thereby increasing the level of glucose transport [33]. When blood levels of insulin are high, skeletal muscle accounts for about 70–80% of total glucose uptake in healthy young adults [34].

Insulin resistance is the decreased ability of a given dose of insulin to increase glucose uptake from an individual’s circulation compared to the normal nondiabetic population [35]. Skeletal muscle insulin resistance is considered the primary defect in T2D [34]. In the prediabetic state, there is resistance to insulin action and an initial compensatory increase in beta-cell insulin secretion to maintain normal plasma glucose levels. This initial compensatory hyperinsulinemia is generally viewed as beneficial in the prediabetic state, but chronic glucose exposure is thought to promote eventual beta-cell failure with a resulting T2D. Although beyond the scope of this review, there is also evidence that hyperinsulinemia could contribute to T2D progression and pathophysiology [36]. It has been suggested that hyperinsulinemia may play a role in promoting oxidative stress in T2D, but further study is needed [37]. In an animal model (Wistar rats), short-term hyperinsulinemia reduces oxidative stress [38].

### 2.1. Skeletal Muscle Insulin Resistance Is a Primary Defect in T2D

Skeletal muscle utilizes both fatty acids and glucose as energy sources to produce ATP. Fatty acids are the primary fuel utilized during rest or mild-intensity exercise, whereas glucose consumption dominates with high-intensity exercise. Under aerobic conditions, pyruvate (from glucose metabolism) can be oxidized by mitochondria, but under anaerobic conditions, pyruvate generates ATP by glycolysis in the cytoplasm. In contrast, fatty acids are utilized only by mitochondria to produce ATP. As mentioned above, skeletal muscle uptake of glucose from the capillary circulation is primarily regulated by the GLUT4 transporter [32]. Exercise and insulin delivery promote glucose transport to muscle but do so through distinct molecular mechanisms that act synergistically [32]. Under hyperglycemic conditions, T2D patients show impaired insulin-stimulated glucose transport into skeletal muscle and decreased intramuscular levels of glucose-6-phosphate (G-6P) compared to normal subjects with the same plasma glucose levels [39,40]. G-6P is a key metabolite for maintaining cellular redox homeostasis [41]. Work by Ciaraldi et al. shows that skeletal muscle GLUT1 expression in T2D is reduced compared to age-matched nondiabetics and that glucose uptake into leg skeletal muscle is reduced in T2D [31].

### 2.2. Skeletal Muscle Glutathione Peroxidase System and Oxidative Stress

As shown in Figure 2, glucose is converted to G-6P by hexokinase (HK) and G-6P is a substrate for glucose-6-phosphate dehydrogenase (G6PD), which is an antioxidant enzyme that reduces nicotinamide adenine dinucleotide phosphate (NADP^+^) to NADPH. This reduction reaction is the first step in the pentose phosphate pathway (PPP) which plays a central role in DM [42]. Intracellular NADPH is essential for reducing oxidized glutathione (GSSG) to reduced glutathione (GSH) which is the main intracellular water-soluble chemical antioxidant. This reaction is catalyzed by GSH reductase (GR). GSH is a tripeptide (gamma-glutamyl-cysteinyl glycine) and a substrate for glutathione peroxidase (GPX), which reduce H_2_O_2_ to water (or lipid hydroperoxides to lipid alcohols) with the formation of oxidized GSH (GSSG) [27,43]. GPX, in concert with GSH, prevents oxidative stress by reducing levels of H_2_O_2_, which is a ROS. The decreased intramuscular level of G-6P seen in T2D is, therefore, a likely contributor to oxidative stress by inhibiting NADPH-dependent GSH recycling. G6PD activity in adult T2D patients is negatively correlated with oxidative stress biomarkers and HbA1c levels [44]. Moreover, G6PD deficiency has been linked with T2D [45]. G6PD deficiency is one of the most common forms of genetic enzyme deficiency.

GPXs are a family of enzymes, with GPX1-4 and GPX6 being selenium-dependent enzymes [43]. GPX1-3 enzymes can reduce H_2_O_2_ as well as small organic hydroperoxides, but GPX4 is uniquely able to reduce structurally complex phospholipid- and cholesterol hydroperoxides [46]. Selenium (Se) is an essential trace element and dietary deficiency of Se results in a marked loss of GPX activity in most tissues [47]. GPX1 is a major antioxidant enzyme found in the cytoplasm and mitochondria of most cells and in vitro studies suggest this enzyme modulates redox-sensitive cellular responses by regulating mitochondrial function [48]. Work by Chung et al. [49] has shown that GPX3 expression in human skeletal muscle mediates the antioxidant effect of peroxisome proliferator-activated receptor-gamma (PPAR-gamma). Moreover, these investigators also show that thiazolidinediones (TZDs), via their activation of PPAR-gamma, induce the expression of skeletal muscle GPX3 and lower H_2_O_2_ levels, which would otherwise cause insulin resistance (see below). TZDs are medications that have been used in the management of T2D. Tissue culture and animal model studies suggest that the GPX family of enzymes participate in numerous regulatory phenomena, but their roles in diabetes and human health require further research [43].

### 2.3. Mitochondrial H_2_O_2_ Emission as a Potential Initiating Event for Insulin Resistance

There is considerable evidence linking mitochondrial oxidative stress and T2D [50]. Pioneering ex vivo, in vivo, in vitro, and dietary experiments by Anderson et al. [16] show that a high-fat diet, either in rodents or humans, markedly increases skeletal muscle mitochondrial H_2_O_2_ emission, subsequently resulting in insulin resistance. Mitochondrial H_2_O_2_ arises from the superoxide dismutase (SOD) catalyzed conversion of the superoxide radical (O_2_^•−^). The early development of insulin resistance in children is strongly associated with obesity and the future development of T2D [51].

We will summarize the work of Anderson et al. in detail, given its etiological significance [16]. These researchers studied mitochondrial H_2_O_2_ emission in isolated skeletal muscle fibers. Mitochondrial H_2_O_2_ emission was increased in skeletal muscle isolated from normal Sprague-Dawley rats fed a high-fat diet compared (for 3 days or 3 weeks) to skeletal muscle from rats fed a standard chow diet (high in carbohydrates). Notably, this effect was prevented by treating the high-fat diet rats with a mitochondrial-specific cell-permeable synthetic antioxidant peptide (SS31). SS31 selectively localizes to the mitochondrial inner membrane and is currently undergoing clinical trials [52]. As outlined in Figure 2, increased emission of H_2_O_2_ into the cytoplasm is expected to result in increased consumption of GSH and formation of GSSG by GPX and a decreased cytosolic GSH/GSSG ratio. This expected shift towards a more oxidized GSH state in the skeletal muscle of the rats fed the high-fat diet was experimentally confirmed by Anderson et al. [16]. Moreover, treatment with SS31 prevented the high-fat diet-induced shift to a decreased GSH/GSSG ratio. Quite remarkably, the normal rats fed a high-fat diet for six weeks developed insulin resistance which was completely blocked by treatment with SS31. Moreover, in rats fed the high-fat diet, glucose ingestion did not result in the activation (by phosphorylation) of AKT (see above) in skeletal muscle as was observed in rats fed the standard-high carbohydrate chow diet. For rats fed the high-fat diet but treated with SS31, AKT phosphorylation was completely restored [16]. Collectively, these data demonstrate that a high-fat diet induces insulin resistance by modulating the AKT signaling pathway and thereby the plasma membrane level of GLUT4.

To establish that increased mitochondrial H_2_O_2_ emission was a primary factor causing insulin resistance, these investigators studied transgenic mice overexpressing human catalase (CAT) in skeletal and cardiac mitochondria (MCAT mice). CAT is normally localized to peroxisomes in most cells and is only found to be present in the mitochondria of rat cardiac tissue [53]. Skeletal muscle mitochondrial H_2_O_2_ emission in the MCAT mice fed a high-fat diet was significantly below that in the control wild-type mice fed the high diet. Most importantly, whole-body insulin sensitivity was preserved in the MCAT mice fed a high-fat diet.

In their final series of ex vivo experiments, Anderson et al. [16] measured mitochondrial H_2_O_2_ emission in human skeletal muscle biopsies from lean, insulin-sensitive males and obese, insulin-resistance male subjects. The obese subjects showed a two- to four-fold increase in mitochondrial H_2_O_2_ emission and a marked decrease (at least 50%) in the GSH/GSSG ratio. Moreover, four hours (the postprandial period) after feeding the lean subjects a high-fat meal, their skeletal muscle mitochondrial H_2_O_2_ emission increased two- to four-fold, and their GSH/GSSG ratio decreased (at least 50%) compared to a 12-h fasting state.

Collectively, these data strongly suggest that a high-fat diet is mechanistically linked to skeletal muscle mitochondrial H_2_O_2_ emission, altered cellular redox status, and the development of insulin resistance. The authors suggest that while both carbohydrate and fat ingestion cause increased mitochondrial H_2_O_2_ emission, the effect of fat ingestion was less transitory and still evident after a 12-h fast. These researchers did not look at plasma POS in either their human or animal models. This would be useful in future studies since it would help determine if plasma POS is produced secondarily by skeletal muscle oxidative stress. Human experiments have shown that plasma oxidative markers increase after acute exercise, suggesting that exercise-induced oxidative stress is not confined to cellular compartments [54].

### 2.4. Hyperglycemia and Glycation Induced Oxidative Stress

While low insulin-stimulated transport of glucose into skeletal muscle is a key etiological factor in T2D, the resulting hyperglycemia is also a fundamental cause of pathophysiology. Hyperglycemia causes an increased glucose uptake in cells relying on insulin-independent GLUT transport, e.g., endothelial cells. Hyperglycemia or high intracellular glucose (or fructose) promotes the non-enzymatic glycation of protein lysine (K) and arginine (R) residues as well as the N-terminal residue. This relatively stable glycation product is termed an Amadori adduct that can further react to form AGEs [55]. The glycation level of the N-terminal residue of hemoglobin (HbA1c) is a diagnostically useful measure of long-term hyperglycemia in DM and for monitoring treatment efficacy. Glycation alters the structure of Hb and its oxygen-binding functions in T2D [56]. Hb is not, however, a unique protein: any protein exposed to high glucose levels can be glycated at susceptible residues. Since proteins are “nanomachines” performing most biological functions, their covalent modification by glucose can be deleterious. K and R residues are at the active site of many enzymes. In addition to proteins, lipids and nucleic acids can also be glycated [57].

The potential functional effects of glycation on proteins relevant to oxidative stress in T2D are not well studied. The effects of glycation on selenium-dependent GPXs would be particularly relevant given their potential etiological roles in T2D, as discussed above. Plasma and RBC GPX activity are lower in T2D compared to healthy controls [58,59]. The lower activity of GPX could be due to dietary selenium deficiency and glycation. The selenium content of plasma GPX is, however, the same for T2D and normal subjects [59]. Moreover, in vitro glycation of GPX with glucose, fructose, or galactose results in loss of GPX activity [59]. Although more research is warranted, these results strongly suggest that glycation of GPXs negatively affects their enzymatic activity. This would be important in GPXs in plasma as well as in the cytoplasm and mitochondria of cells where glucose uptake is not insulin sensitive (e.g., RBCs and endothelial cells). Work by Goyal et al. also indicates that RBC GPX activity level is more suppressed in obese T2D compared to non-obese T2D, suggesting that obesity is a contributor to oxidative stress [58].

### 2.5. Advanced Glycation End Products (AGEs) and Oxidative Stress in T2D

The Amadori products resulting from protein glycation can undergo further covalent modifications to form protein-AGEs [60]. As reviewed by Nowotny et al. [60], oxidative stress is thought to promote the formation of protein-AGEs which, in turn, can act to promote additional oxidative stress. The role of AGEs in promoting oxidative stress and in the development of T2D is an area of active research [61]. AGEs bind to an AGE-specific receptor (AGER or also termed RAGE) that, in turn, activates the transcription factor nuclear factor NF-kappa-B (NFKB), which promotes the expression of inflammatory genes [62]. AGEs are thought to contribute to chronic oxidative stress in T2D and negatively impact antioxidant systems [60]. Plasma proteins exposed to high levels of glycating sugars are, therefore, subject to glycation and AGE formation. Albumin, the most abundant plasma protein, forms site-specific AGE products that are high in patients with T2D and poor glycemic control [63].

### 2.6. AGEs, Skeletal Muscle GLUT4 Expression, and Insulin Resistance

Interesting in vitro and in vivo work by Pinto-Junior et al. [62] in an animal model indicates that albumin-AGE, in addition to being a biomarker for poor glycemic control, could also induce insulin resistance by decreasing the expression of the GLUT4 gene (Scl2a4) in skeletal muscle. In this work, normal rats treated with albumin-AGE for 12 weeks developed insulin resistance as well as a decreased Scl2a4 mRNA expression and GLUT4 protein content. These data suggest that AGEs, independent of hyperglycemia, can affect glucose uptake into skeletal muscle by decreasing the expression of GLUT4. This study further suggests the possibility that repeated bouts of transient postprandial hyperglycemia could gradually induce the formation of sufficient levels of albumin-AGE to induce a more permanent long-term insulin resistance, as can be found in pre-diabetes. Rammos et al. [64] reviewed the evidence linking chronic hyperglycemia-induced oxidative stress to T2D with an emphasis on endothelial cell dysfunction.

### 2.7. Vascular Endothelial Cells, Oxidative Stress, and T2D

Vascular endothelial cells are in direct contact with the elevated plasma glucose levels characteristically found in poorly controlled T2D. As stated by Kida et al. [65] “hyperglycemia is the primary cause of vascular complications occurring in individuals with diabetes”. Endothelial cells primarily rely on GLUT1 (an insulin-independent transporter) for glucose transport. In contrast to skeletal muscle cells, intracellular glucose levels in endothelial cells markedly increase in response to hyperglycemia and accumulate G-6P [66,67]. Endothelial cells utilize the polyol pathway to remove excess intracellular glucose [68,69].

Under hyperglycemic conditions, about 30% of blood glucose moves through the polyol pathway [69]. In this pathway (see Figure 3), glucose is first reduced to sorbitol by aldose reductase and this reaction consumes NADPH. In the second step, sorbitol is converted into fructose by sorbitol dehydrogenase and this reaction consumes NAD^+^ producing NADH. The net result of these two steps is the conversion of NADPH to NADH and increased intracellular levels of fructose. Decreased intracellular NADPH could result in a decreased ability of GR to reduce GSSG to GSH and thereby reduce the ability of GPX to reduce H_2_O_2_ or lipid hydroperoxides with a resulting increase in oxidative stress [69,70]. Moreover, fructose is a reducing sugar and, like glucose, can glycate proteins, alter their functions, and result in protein-AGEs. In vitro studies indicate that fructose forms both glycation products and AGEs more effectively and rapidly than glucose [71]. Animal models also suggest that fructose promotes oxidative stress to a greater extent than glucose [72]. Dietary AGEs are formed during high-temperature food processing, as is typical for the Western diet [18]. Individuals with T2D show increased plasma POS, and macro-and microvascular endothelial dysfunction, after consuming a meal rich in AGEs [19].

## 3. System Medicine and a Future Redoxomics Approach to Pediatric T2D

Historically, medicine has largely utilized a reductionist approach in which disease states are generally reduced to a single organ or defect. Consequently, this approach often overlooks potential interactions between both intrinsic and extrinsic modulators and environmental risk factors [73]. Alternatively, an integrative systems medicine approach is becoming increasingly favored. Redoxomics is a branch of systems medicine focusing on oxidative stress, reactive oxygen species, and antioxidants. Systems medicine, also referred to as precision or “P4” medicine, captures the power of omics technologies, such as genomics, epigenomics, proteomics, and metabolomics, and their interaction with environmental factors like nutrition and the gut microbiome [74,75,76,77,78]. The goal of P4 medicine is to provide an integrated healthcare approach that is “predictive, preventive, personalized and participatory” [79,80]. In short, systems medicine is interdisciplinary and utilizes omic data to reveal pathophysiological processes useful for predicting the inherent risk of an individual patient to developing a disease such as diabetes and to assess the success likelihood of specific treatments [74,75,76,77].

As detailed above, considerable evidence supports the role of oxidative stress as an initial etiological factor for pediatric T2D. It follows that redoxomics would be of optimal clinical benefit while simultaneously providing detailed molecular insights into pathophysiology. A comprehensive application of redoxomics to either adult or youth-onset T2D is currently lacking. An early childhood study with multiple follow-up periods would be ideal since it might identify very early redoxomics biomarkers providing precision guidance for preventing or slowing future T2D development before irreversible pathology occurs. We will next introduce the potential of a generalized systems medicine approach to T2D and detail the ongoing contributions of omics technology to risk assessment, patient stratification, and predicting drug responsiveness. We will also make a note of existing omics data relevant to oxidative stress and T2D since this would help guide future redoxomic studies. These omics studies have mostly been limited to adults of Western European origin, with some newer additions from East Asian populations [81,82,83]. Genomic technology is quite advanced, very comprehensive, cost-effective, and has overcome many regulatory hurdles for providing data analysis reports directly to adult consumers.

### 3.1. Genomics and T2D

T2D has a strong genetic predisposition and published genome-wide association studies (GWAS) have identified over 500 single nucleotide polymorphisms (SNPs) associated with susceptibility [81]. Nevertheless, these genetic variants have accounted for only about 10–20% of the heritability of T2D and some of these variants are also associated with obesity [83]. The relevant genetic risk factors can be combined into a polygenic T2D risk score with the potential to inform healthcare decisions, e.g., the potential to respond to an oral diabetic medication [84]. Very cost-effective commercial genotyping can currently provide an estimate of T2D risk development for adults. The 23andme company provides a T2D polygenic sore based on over 1000 genetic variants. About 20% of their research population has a T2D genetic risk equal to that of being overweight. Nevertheless, 23andme did not obtain Food and Drug Agency (FDA) clearance for the genetic risk score as it was characterized as a wellness product that does not make a diagnosis or provide medical advice. A genetic risk score for youth onset T2D has not been published but could potentially provide a sign to initiate early and robust lifestyle interventions.

#### 3.1.1. Genomics and Stratification of Adult T2D Patients

A second major contribution of genomics to T2D lies in its potential to help categorize diabetes types and thereby provide more individualized healthcare. A controversial Swedish study recently proposed that adult diabetes be subdivided into five groups rather than the current two [85]. Six measures were utilized to assign a patient to a group: body BMI, age at diabetes diagnosis, HbA1C level, beta-cell functionality, insulin resistance, and the presence of diabetes-related autoantibodies. Significantly, the five types of DM categorized by these parameters were genetically distinct. These are very promising results and extending these studies to additional ethnic populations as well as the pediatric population could have enormous healthcare-related significance.

#### 3.1.2. Genomics May Help Stratify Youth Onset T2D Patients

The possibility of using genomics to help stratify youth onset T2D is a high-priority future goal. In a consensus report, Nadeau et al. [86] reviewed the clinical evidence suggesting that youth-onset T2D is unique and distinct from T2D in adults. Youth-onset T2D is characterized by both a rapid beta-cell decline and an accelerated progression. The underlying pathophysiological mechanisms for these unique alterations are not well understood. The consensus report asserts that new and comprehensive strategies for treating youth-onset T2D “are urgently needed” [86]. A systems medicine approach will most likely contribute to future efforts at developing targeted treatments.

#### 3.1.3. Genomics and Oxidative Stress

GWAS studies on oxidative stress in humans are still “a work in progress,” although studies in other organisms have provided a useful framework [87]. Variants in CAT, GPX4, and GSR (glutathione reductase) and the transcription factor nuclear factor-erythroid factor 2-related factor 2 (Nrf2) have been found to modulate human oxidative stress [88]. Despite their potential importance, biomarkers for oxidative stress in T2D are not clinically used to categorize DM or guide healthcare. Tabatabaei-Malazy et al. have reviewed the potential relevance of polymorphisms in antioxidant genes to T2D or its clinical complications [89]. These authors stress the need for additional studies focusing on epigenetic mechanisms regulating the expression of antioxidant enzymes. Future GWAS studies in the pediatric population could benefit from looking at the genetic variants contributing to POS, the development of pediatric insulin resistance, prediabetes, and T2D. Given the potential importance of mitochondria in T2D, it may prove useful to place more emphasis on mitochondrial DNA variants, as suggested by Wang et al. [90]. Mitochondrial DNA is particularly susceptible to oxidative stress-induced mutations and DNA damage [90]. Circulating mitochondrial DNA, thought to arise from stressed cells, may be a non-invasive biomarker for T2D [91].

### 3.2. Metabolomics, Oxidative Stress, and T2D

The potential clinical applications of metabolomics to oxidative stress-induced alterations in cytosolic, mitochondrial, and redox metabolites hold great promise [92]. It has been well documented, for example, that branched-chain amino acids (BCAAs) are elevated in T2D [93]. BCAAs include leucine, isoleucine and valine. In vitro and ex vivo experiments demonstrate that BCAAs promote endothelial dysfunction via increased mitochondrial ROS generation and increased formation of peroxynitrite and 3-nitro-tyrosine [94]. The induction of oxidative stress in endothelial cells is thought to be caused by activation of the mTORC1 pathway [94]. GWAS/metabolomic studies have confirmed a causal role between the level of insulin resistance and levels of circulating BCAAs [95].

In 2011, Wang et al. utilized a targeted metabolomic approach to evaluate the risk of developing T2D in adults [96]. In this study, a metabolic profile (over 60 metabolites) was measured in the fasting plasma of 2422 healthy, non-diabetic subjects. After a 12-year follow-up period, 201 individuals developed T2D. Quite significantly, plasma BCAAs levels (and that of two aromatic amino acids) had a significant association with future T2D development that was superior to BMI, dietary pattern, and fasting glucose [96].

Lipidomics is a subset of metabolomics looking only at lipid metabolites in a given biofluid/biosample. When fasting plasma is used, the lipids sampled will primarily represent lipids associated with lipoproteins and not cellular biomembranes. Recent research in a Swedish population shows that a “lipidomic risk,” based on a single mass spectrometric measurement in a plasma sample, can identify a subset of adult individuals at high risk for developing T2D. Moreover, the lipidomic risk score was largely independent of the polygenic risk score [97].

### 3.3. Proteomics and T2D

Chen and Gerszten have reviewed the potential of proteomics for advancing T2D healthcare [98]. These authors suggest that integrating circulating metabolomics and proteomics (i.e., multiomics) would be an optimum strategy. For large-scale population studies and future pediatric studies, the emphasis on plasma is realistic. As postulated above, skeletal muscle mitochondrial alterations are likely to be of etiological significance for T2D, and plasma (or RBC) would not be as mechanistically informative as a skeletal muscle biopsy. Future mechanistic multiomic studies with skeletal muscle biopsies remain critical for advancing an in-depth understanding of T2D. For children, skeletal muscle biopsies are bioethically justified only with specific medical justification. Following tissue damage, tissue-specific proteins can leak into plasma but are usually present only in low abundance making their detection and quantification technologically challenging (more on this below).

#### 3.3.1. Protein Glycation and Oxidation in T2D

Plasma and RBC samples can provide quantitative information on proteins that have been modified by glycation, AGE formation, or oxidative stress. A comprehensive list of glycated proteins from the plasma and RBCs of control and T2D subjects has been published, with several proteins being significantly more glycated in T2D subjects [55]. Interestingly, GPX3 (plasma glutathione peroxidase) was found to be glycated at four distinct peptide sequences. Carbonylated plasma proteins can be formed directly from ROS modifications of susceptible amino acid residues or by the covalent addition of adducts with carbonyl groups as they arise from glycation [99,100]. Carbonylated proteins are not, therefore, a good measure of ROS-mediated protein oxidation alone. Although lacking quantification, Bollineni et al. [100] looked at the presence of carbonylated plasma proteins in a small population of lean subjects and obese subjects with or without T2D. A unique set of carbonylated proteins were found only in the obese T2D subjects. These unique proteins need to be confirmed by additional studies (including pediatric subjects) but strongly suggest that carbonylated plasma proteins could provide biomarkers for obesity-induced T2D.

#### 3.3.2. The Proximity Ligation Assay, T2D, and Metformin Response Stratification

Traditional proteomics utilizing highly specialized modern spectrometry equipment is well suited for basic research, exploratory clinical research, and proteome discovery. Fu et al. [101] have detailed the potential power of label-free quantitation for biomarker discovery in diabetes. The utility of the proximity ligation assay (PIA) for T2D precision medicine and biomarker discovery has recently been demonstrated in a modest cohort of healthy and T2D subjects [102].

PIA can be used for the detection of proteins, post-translational modifications (e.g., glycation) and protein-protein interactions [103,104]. In this assay, two complementary DNA-tagged antibodies bind to epitopes in close proximity to each other on the same target protein. DNA in the matched pair of tagged antibodies hybridize, is extended by DNA polymerase, and amplified by the quantitative polymerase chain reaction (qPCR) or next-generation sequencing (NGS). The amplified DNA is essentially a unique double-stranded “barcode” for a specific target protein and the number of readouts of that unique barcode is a quantitative measure of the target protein’s level in the sample. This assay can be multiplexed to quantify many proteins in a given sample.

Utilizing microliter amounts of fasting plasma, Zhong et al. could quantify about 1500 proteins (with NGS readout) in an adult Swedish population, including low abundance proteins [102]. They found that the plasma proteome significantly varied from one healthy individual to the next, but each had a unique profile that was stable over time. Two populations of T2D subjects were studied, i.e., obese and non-obese. Four proteins were found to have significantly altered expression between these two groups (one was leptin). They then examined the protein profiles of non-obese T2D patients with non-obese healthy patients and found 32 proteins with altered expression. The authors suggest these 32 proteins might serve to identify subjects at increased T2D risk despite not being obese. They next attempted to stratify T2D patients based on their responses to metformin treatment. A total of 40 T2D subjects were treated with metformin for three months and these were stratified into responders, non-responders, and an intermediate group. Significantly, 30 proteins in the plasma proteome measured at baseline (before the start of metformin intervention) could distinguish the responders from the non-responders. This result suggests that the panel of 30 proteins could be used to predict whether a Swedish adult T2D patient would respond well to metformin therapy.

In contrast to the overall positive responses of T2D adults to metformin alone or insulin treatment followed by metformin, youth-onset T2D shows no improvement in progressive beta-cell deterioration [105]. This again points to the uniqueness of youth-onset T2D and the speculation that redoxomics could help define relevant factors for preserving beta-cell function and preventing T2D progression.

Once a subset of proteins has been established to have clinical relevance, proteomics can shift from a discovery mode to a targeted mode in which only this subset is quantified. To be of practical use in systems medicine, targeted proteomic analyses need to be highly sensitive and specific with high throughput and long-term reproducibility. The PEA assay is very promising in this regard and could easily be expanded to look at relevant plasma-proteins covalently modified by glycation, AGE formation, and oxidative stress.

## 4. System Medicine, Nutrigenomics and T2D

As detailed above, obesity, excess dietary calories, lack of exercise, and poor diet all contribute to T2D risk and progression (see Figure 1). Interventions aimed at correcting these lifestyle issues are central to individual care and relevant public health policies and programs. The American Diabetes Association supports “evidence-based nutrition standards for school meals and snacks, and other child nutrition programs” [106]. In addition to total calories, fat levels and type, fiber content, and glycemic index, it would also be useful to consider the levels of various dietary antioxidants and dietary AGEs [107]. Nutrients and dietary patterns are also known to influence gene expression in ways potentially relevant to the causes and development of T2D [108].

Nutrigenomics is emerging as an area possibly important in T2D prevention and treatment [109]. Nutrigenomics is a branch of systems medicine studying the interactions between genes and nutrients [110]. Many polyphenols, in addition to having intrinsic antioxidant properties, are also bioactive compounds that can potentially alter the expression of genes important to T2D [109]. Resveratrol, quercetin, genistein, catechins, and curcumin are bioactive dietary polyphenols that may play a role in T2D prevention/management [109,111,112]. Both GSH and ascorbate are water-soluble antioxidants that may also play protective roles in T2D. Dietary ascorbate supplementation may improve glycemic control, but further clinical trials are necessary [113]. Similarly, a short-term pilot study with glycine and N-acetylcysteine supplementation (precursors of GSH) improves mitochondrial dysfunction and insulin resistance in adult T2D patients [114,115].

## 5. Conclusions

Adopting a systems medicine approach to T2D may require significant clinical and educational changes. Some medical schools, like Georgetown University, already implement systems medicine into their curriculum [75,116]. It is important to note that much of the preliminary research into the systems medicine of diabetes has been done with adult populations of mostly Western European descent. There is a critical need to extend these studies to pediatric populations and ethnically diverse populations. Youth onset T2D may prove to have a genetic risk score with a unique set of gene variants compared to adult-onset T2D. As detailed above, dietary-induced oxidative stress (see Figure 1) is likely an early initiating event for T2D progression and, therefore, of etiological significance to pediatric T2D. The comprehensive characterization of redox status provided by redoxomics holds great future promise in helping to evaluate the efficacy of lifestyle and nutritional interventions for T2D as well as the application of precision medicine. A multiomics approach that would utilize a targeted panel of genomic, metabolomic, and proteomic biomarkers is likely to be optimal.

## Figures and Tables

**Figure 1 antioxidants-11-01336-f001:**
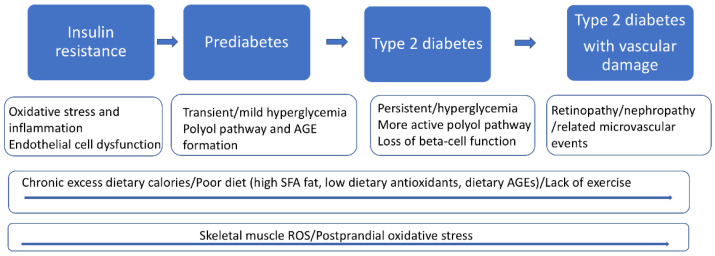
The stages of Type 2 Diabetes (T2D). Chronic excess calories produce skeletal muscle mitochondria reactive oxygen species (ROS) emission, which, along with dietary AGEs, initiates insulin resistance and contributes to postprandial oxidative stress. Insulin resistance and the resulting hyperglycemia activate the polyol pathway with additional AGE-formation that amplifies oxidative stress/inflammation and contributes to beta-cell dysfunctions. Skeletal muscle ROS formation and postprandial oxidative stress will continue to act deleteriously throughout all four stages of T2D (long blue arrow) unless treated with lifestyle and medical interventions. High levels of dietary AGEs and low levels of dietary antioxidants will also contribute to postprandial oxidative stress and accelerate T2D progression.

**Figure 2 antioxidants-11-01336-f002:**
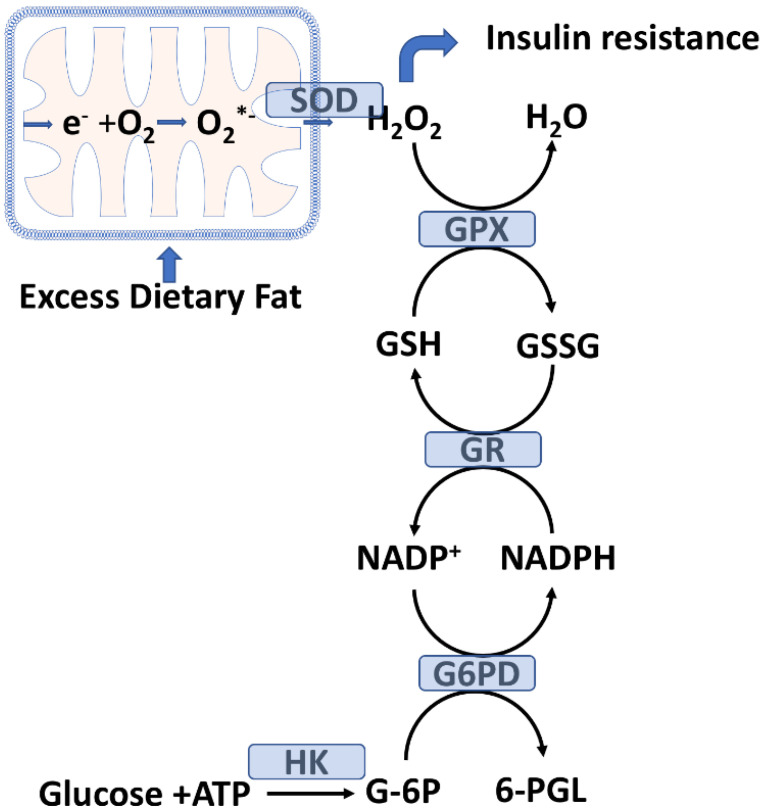
A model linking excess dietary fat to skeletal muscle mitochondrial emission of H_2_O_2_, insulin resistance, and the glutathione peroxides system. Excess dietary fat promotes electron leakage and the production of superoxide (O_2_^•−^) anion radicals. Superoxide dismutase (SOD) converts O_2_^•−^ to H_2_O_2_, which is reduced by glutathione peroxidase (GPX) with the consumption of reduced glutathione (GSH) and the formation of oxidized glutathione (GSSG). GSSG is recycled back to GSH by glutathione reductase (GR) with the consumption of NADPH and the formation of NADP^+^. Glucose-6-phosphate dehydrogenase (G6PD), in turn, catalyzes the reduction of NADP^+^ to NADPH utilizing glucose-6-phosphate (G-6P) provided by the action of hexokinase (HK) on glucose. Excess H_2_O_2_ emission into the skeletal muscle cytoplasm has been linked to insulin resistance (see text) and a shift to reduced levels of GSH and increased levels of GSSG.

**Figure 3 antioxidants-11-01336-f003:**
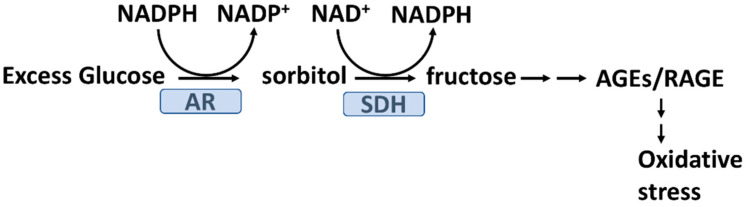
The polyol pathway and oxidative stress. For cells in which hyperglycemia produces high intracellular glucose levels, the polyol pathway is activated. Excess intracellular glucose is converted to sorbitol by the action of aldose reductase (AR), which consumes NADPH. NADPH is required to recycle oxidized glutathione (GSSG) to reduce glutathione (GSH) and reduce peroxides (see Figure 1), which would otherwise cause oxidative stress. Sorbitol is converted to fructose by sorbitol (SDH). Fructose is an effective glycating sugar-forming AGEs that promote oxidative stress and inflammation.

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
