# Peer review of "Oxidative Stress in Type 2 Diabetes: The Case for Future Pediatric Redoxomics Studies"

_antioxidants, 2022, doi:10.3390/antiox11071336_

Round 1
Reviewer 1 Report
The review by Alu and colleauges describes describes several areas of redox biology that are already well characterized. In the first part, they discuss in length the studies by Neufer as well as Ethan Anderson, showing that skeletal muscle mitochondria H2O2 emission as a ROS signal and its effects on insulin resistance. This has been described many times and well known for about a decade now. The authors do not detail how this would connect to pediatric diabetes which makes the title misleading. At minimum, I would recommend removing the word ‘pediatric’ from the title.
There are several enzymes that can counteract H2O2 especially in the cytoplasm, much more so than superoxide. So to say that mitochondrial H2O2 is an initiating event for insulin resistance is overstated.
The authors also mention hyperglycemia induced oxidative stress and discuss AGE (advanced glycation end products) and the role of the polyol pathway in generating AGEs. Again, very well known topics already and not sure what the novel view or new addition would be from this review.
Next, they talk about “redoxomics approach”. This is where I got confused a bit. It is a bit nebulous and I am not sure this would move the field of pediatrics forward.
For example, consider the quote: “Redoxomics is a branch of systems medicine focusing on oxidative stress ROS and antioxidants”. How would this whole antioxidant concept work, even if we identify new compounds? It has failed the past 50 years because the field never took compartmentalization or kinetics into consideration. The umbrella approach of giving antioxidants will likely not solve pediatric or adult T2DM.
In the Summary, the authors suggest perhaps that some of these antioxidants alter gene expression of certain genes and that could be more significant than just the antioxidant properties. If this is true, that would be great but I do not see how antioxidants would lead to this. Supplements would be picomolar or even less in most systems. Evolution gave us 10 mM GSH and 100 micromolar ascorbate – thus it is hard to compete with those two using antioxidants. The authors did not address this point.
Additional points to consider: the section that talks about protein glycation –it is hard to discern if you got more protein carbonyls, whether indeed you have more ROS in T2DM or simply you have less protein turnover of damaged or modified proteins. You can get more protein carbonyl both ways, so protein carbonyls are not a good marker of oxidative stress. The authors did not address this point.
Sometimes they refer to GPX enzymes as H2O2 removers – yes that is true mostly for GPX1, the others especially GPX4 is a lot more targeted towards lipid hydroperoxides, especially GPX4 since it is the only one that removes phospholipid hydroperoxides. The authors did not address this point.
Reviewer 2 Report
Major Comments
1. Abstract section misses structure and does not end with clear conclusions.
2. Paper misses a good design, structure and message. After reading this paper it is not clear why redoxomics studies would be (especially) helpful to study oxidative stress in children with type 2 diabetes.
3. Page 2, lines 59-61: The authors state that “systems medicine approach, particularly redoxomics may provide optimal risk assessment and help guide healthcare strategies. Is this pure speculation of the authors? The authors should present/provide (better) arguments/facts to support this statement. Is studying redoxomics useful to assess the risk for type 2 diabetes and/or its complications?
4. In which respect can redoxomics studies in children provide other information than in adults? Is there evidence that the role of oxidative stress as initial etiological factor for T2D differs between children, adolescents, and adults?
5. As the authors state: “Redoxomics is a branch of systems medicine focusing on “omic” data related to antioxidants, oxidative stress and reactive oxygen species. However, the paper is not focused on the role of redoxomics, but also extensively discusses other branches of systems medicine genomics, epigenomics, lipidomics, proteomics, and metabolomics and their theoretical background. However, it is unclear why this would especially be important when studying oxidative stress in children with Type 2 Diabetes. Why is study of genomics, metabolomics and proteomics relevant for oxidative stress in type 2 diabetes and redoxomics studies?
6. Type 1 diabetes most often occurs in children while the incidence and prevalence of type 2 diabetes mellitus in children and adolescents are considerably lower. It is unclear from the manuscript why measurement of plasma POS parameters is more important for type 2 diabetes than for type 1 diabetes especially since the authors suggested that hyperglycemia is a fundamental cause of pathophysiology (“hyperglycemia is the primary cause of vascular complications occurring in individuals with diabetes.”).
7. Prospective analyses of a Pima Indian cohort have identified an independent role for basal hyperinsulinemia in the development of diabetes. The authors do not discuss whether hyperinsulinemia plays a role in oxidative stress in type 2 diabetes.
8. Page 9, lines 394-399. What is the intention of the authors to discuss results of a not peer-reviewed paper?
9. In the conclusions (page 12) the authors introduce for the first time nutrigenomic and some other topics which were not previously discussed in the paper. However, in the conclusions section the authors should only briefly summarizing the overall findings/arguments but not introduce new ideas/findings/arguments
Reviewer 3 Report
This is a well-documented and well-written review addressed to the role of redoxomics in the natural history of T2D in children/adolescents. The authors present a clear picture of critical pathogenetic changes in relation with oxidative stress and the potential contribution of various "omics" techniques in understanding pathogenesis of T2D.
Round 2
Reviewer 1 Report
While I cannot see the response to Reviewer 2 that the authors refer to in their response to Reviewer 1, the other comments were addressed.
Reviewer 2 Report
The paper is sufficiently improved, but some paragraphs are still difficult-to-follow and to-understand.